# California condor microbiomes: Bacterial variety and functional properties in captive-bred individuals

**Lindsey Jacobs[1], Benjamin H. McMahon[1], Joel Berendzen[2], Jonathan Longmire[1], Cheryl Gleasner[1], Nicolas W. Hengartner[1], Momchilo Vuyisich[3], Judith R. Cohn[1], Marti Jenkins[4], Andrew W. Bartlow[1], Jeanne M. Fair[1]***

**1** Los Alamos National Laboratory, Los Alamos, New Mexico, United States of America, **2** GenerisBio, Santa Fe, New Mexico, United States of America, **3** Viome, Los Alamos, New Mexico, United States of America, **4** The Peregrine Fund, Boise, Idaho, United States of America

* jmfair@lanl.gov

**Data Availability Statement:** Sequence data files are deposited in the sequence read archive under bioproject number PRJNA297579. The source code for sequedex is available at https://github.

## Abstract

Around the world, scavenging birds such as vultures and condors have been experiencing drastic population declines. Scavenging birds have a distinct digestive process to deal with higher amounts of bacteria in their primary diet of carcasses in varying levels of decay. These observations motivate us to present an analysis of captive and healthy California condor (*Gymnogyps californianus*) microbiomes to characterize a population raised together under similar conditions. Shotgun metagenomic DNA sequences were analyzed from fecal and cloacal samples of captive birds. Classification of shotgun DNA sequence data with peptide signatures using the Sequedex package provided both phylogenetic and functional profiles, as well as individually annotated reads for targeted confirmatory analysis. We observed bacterial species previously associated with birds and gut microbiomes, including both virulent and opportunistic pathogens such as *Clostridium perfringens*, *Propionibacterium acnes*, *Shigella flexneri*, and *Fusobacterium mortiferum*, common flora such as *Lactobacillus johnsonii*, *Lactobacillus ruminus*, and *Bacteroides vulgatus*, and mucosal microbes such as *Delftia acidovorans*, *Stenotrophomonas maltophilia*, and *Corynebacterium falsnii*. Classification using shotgun metagenomic reads from phylogenetic marker genes was consistent with, and more specific than, analysis based on 16S rDNA data. Classification of samples based on either phylogenetic or functional profiles of genomic fragments differentiated three types of samples: fecal, mature cloacal and immature cloacal, with immature birds having approximately 40% higher diversity of microbes.

## Introduction

The California condor (*Gymnogyps californianus*) is a species that dramatically declined during the 20th century from poaching, lead poisoning, and changes in habitat [1]. In 1987, the last remaining individuals were captured from the wild to take part in a breeding program. This breeding program was successful, and in 1992 a small number of condors were

com/lanl/sequedex-core, while an executable release, complete with data modules, is available at https://github.com/Sequedex/Sequedex-build/releases/tag/v2.r20181211. Documentation for the sequedex package is available at https://sequedex.readthedocs.io/en/latest/.

**Funding:** The authors thank Los Alamos National Laboratory's Laboratory Directed Research and Development Program and the Center for Space and Earth Science for the sole funding of the entirety of this study. Triad National Security, LLC, is the operator of the Los Alamos National Laboratory (LANL) under contract no. 89233218CNA000001 with the US Department of Energy. All authors, except MJ at Peregrine Fund, which was not funded, were funded by Los Alamos National Laboratory. MV and JB were at Los Alamos National Laboratory for all parts of their contribution and have current addresses included for their new work organizations. The Los Alamos National Laboratory funder provided support in the form of salaries for all authors, except MJ at the Peregrine Fund, but did not have any additional role in the study design, data collection and analysis, decision to publish, or preparation of the manuscript. The specific roles of all authors are articulated in the 'author contributions' section.

**Competing interests:** The authors declare that there is no conflict of interest regarding the publication of this paper. All work was completed and funded at Los Alamos National Laboratory. Two authors (MV and JB) have since left Los Alamos National Laboratory and their new commercial affiliations and this does not alter our adherence to PLOS ONE policies on sharing data and materials.

reintroduced into the wild. Today there are over 500 California condors living in the wild or in captivity. However, there is less genetic diversity in today's California condors because the 20th century dramatic decline in condor numbers also resulted in an 80% reduction of genetic diversity [2]. This lack of genetic diversity could have health effects, effects which could be enhanced by the potentially pathogenic infectious diseases California condors are exposed to in the wild [3]. Due to these potential impactors on condor health, indicators of condor health both in captivity and in the wild are of great interest to assess both population and individual health and condition.

The microbiome (the collective genome of microbial species in a sample) may be a good indicator of health and condition in avian species [4]. Gut microbiota influence the health and physiology of vertebrates, indicating stressors faced by condors could be assessed using gut microbiota [5]. For example, lower nestling body condition in Great Tits (*Parus major)* is correlated with large shifts in microbial diversity and composition [6]. Metagenomic profiling of microbial communities has already been applied to provide both phylogenetic and functional information for environmental samples [7], wildlife [8], domestic animals [9], and endangered species [10, 11]. Thorough reviews of the avian microbiome exist and could be used in conservation efforts for avian species such as the California condor [4, 5, 12]. In conservation efforts for any species, the microbiome of healthy animals in captivity will provide a valuable resource for comparison with microbial data for wild individuals of the same species [13].

The avian lifestyle is unique from that of many animals, suggesting the role of bacteria in the avian microbiome may be different from what is typically seen. This is particularly true for scavenging birds such as vultures, in which near-neighbor human pathogens have been found in abundance [14]. It is thought that vultures tolerate these potentially pathogenic bacteria in order to benefit from the bacterial breakdown of carrion [15, 16]. To safely digest non-beneficial bacteria ingested from their primary diet of carcasses in varying levels of decay, scavenging birds such as vultures and condors have an extremely acidic digestive system (1.9 pH in vultures compared to 2.9 pH in humans) [17, 18]. Signatures of health in scavenger birds may be significantly different from other bird species due to this digestive adaptation, and analysis of the microbiome in representative healthy scavenger birds will be necessary for useful comparison with wild individuals.

However, recent studies increasingly note the effect of captivity on the microbiome of animals [19, 20]. This is particularly true if the gut microbiome may be impacted by changes in diet from wild counterparts [21]. In addition, the importance of studying the microbiome in wildlife from an evolutionary standpoint has been highlighted [22]. Despite these issues, and prior to samples from wild condors being available, a baseline of variability and diversity of microbiome communities in captive condors should be assessed. In birds, the microbiome of growing chicks and nestlings have been shown to both have increased and decreased microbial diversity [6, 23–26]. These findings emphasize the importance of finding a baseline for California condors.

Advances in signature-based analysis of genomic data demonstrate an ability to rapidly extract both phylogenetic and functional information from shotgun metagenomics data [12]. Compared to the 16S rDNA sequencing approach, shotgun metagenomics allows for the analysis of bacterial diversity in samples, yet also provides information on the presence of fungal, parasitic, and DNA virus pathogens [27]. In addition, sequencing the 16S rDNA gene, while providing an economical and well-understood technique to survey the phylogenetic composition of microbial communities, does not provide information regarding metabolic pathways or antimicrobial resistance patterns. Shotgun metagenomic sequencing promises to identify resistance genes, their point mutations, and other gene functional changes that can identify events such as exposure to lead contaminants. In this work, we apply the Sequedex

metagenomics analysis package [28] to demonstrate the analyses of these features on micro-biome data from California condors. Furthermore, we compare 16S rDNA and shotgun meta-genomics sequencing in metrics of microbial composition and microbial diversity.

The aims of our study are twofold. The first objective is to describe the microbiome in healthy California condors of different ages in captivity to characterize individual variation in the microbial community structure and bacterial genes under a similar environment. Captive individuals of condors of different ages offer an opportunity to test whether the diversity of microbial communities increases or decreases with age. Our second objective is to compare the sequencing approach of 16S rDNA with shotgun metagenomic sequencing for the condor microbiome. Data from captive California condors can be used as a baseline for future comparison of microbiome variation and diversity with that of wild California condors.

## Results

Twenty species of bacteria comprised 89% of the bacterial identifications in the captive healthy California condor microbiomes (Table 1). All of these bacteria had previously been associated with birds and/or gut microbiomes (For example, see Ballou 2016, Larsen 2015, Prabhakar 2012, Vela 2015, and Vital 2014). For most samples, ~30,000 rDNA reads were obtained, while the shotgun sequencing yield was 10–15 million reads per sample. Between 4–58% of the meta-genomic shotgun reads were identified as bacterial when analyzed with Sequedex, with 40% typical for the fecal samples. There was more dispersion in the percent of recognized reads in cloacal samples due to the variability in the amount of condor DNA present. Condor DNA contains a much smaller fraction of coding DNA than bacteria, and thus has a smaller fraction of reads that were recognized by the peptide signatures in Sequedex.

### Microbial composition

As described in the Sequedex documentation (https://sequedex.readthedocs.io/en/latest/) and shown in S1 Fig, each read recognized by Sequedex is placed at a node of the phylogenetic tree, according to the signature peptides contained in that read. The patterns of assigned reads shown are typical of those of single species, with the reads concentrated in a line from the root of the tree towards the leaves. This pattern arises because 151 base pair reads contain only a limited amount of phylogenetic information, with some reads specifically identified near the leaves of the tree and others that more ambiguously identify the sample's organisms. All of these reads should, however, be counted to estimate the relative abundance of the different phylogenetic categories found in the sample.

S1 Fig shows Sequedex's phylogenetic assignment of reads for abundant taxa in four regions of the tree, with over 100,000 reads assigned in each of the examples provided. Each of the examples shown is consistent with the presence of a single species in that area of the phyloge-netic tree, with the *Lactobacillus* example typical of a close match to two closely related species in the reference database, and the *Clostridium sp.* example consistent with a close match to a reference organism (*C. perfringens* in this case) that is distinct from other organisms in the reference database. Both the *Fusobacterium* and *Propionibacterium* examples are more ambiguous, reflecting either a mixture of related organisms or a novel organism with a genome inventory acquired from several of the reference organisms.

The within-sample Simpson's diversity based on reads rolled up into phyla significantly differed among the three groups (ANOVA: $F_{2,27} = 19.41$, $P < 0.001$; Fig 1) and was highest in the cloacal samples of the younger birds ('Cloacal, immature', Fig 1). Cloacal mature had the lowest diversity. Post-hoc tests show that all three groups significantly differ from each other (Tukey post-hoc test: all p-values $< 0.05$). In the immature cloacal samples, a variety of

**Table 1. Attributes of collected fecal and cloacal samples.** Both 16S and DNA shotgun metagenomic sequencing were performed on most samples. The two most-prevalent species are indicated for each individual in Table 1. Dominant species ascribed by the DNA shotgun metagenomics: *Cje* = *Corynebacterium jeikeium*, *Pac* = *Propionibacterium acnes*, *Sfl* = *Shigella flexneri*, *Cpe* = *Clostridium perfringens*, *Fva* = *Fusobacterium varium*, *Bov* = *Bacteroides ovatus*, *Ckr* = *Corynebacterium kroppenstedtii*, *Eco* = *Escherichia coli*, *Sps* = *Staphylococcus pseudintermedius*, *Pav* = *Propionibacterium avidum*, *Dac* = *Delftia acidovorans*, *Cul* = *Corynebacterium ulcerans*, *Bsu* = *Brevundimonas subvibrioides*, *Esp* = *Enterococcus sp.*, *Sma* = *Stenotrophomonas maltophilia*, *Ljo* = *Lactobacillus johnsonii*, *Fmo* = *Fusobacterium mortiferum*, *Pan* = *Peptostreptococcus anaerobius*, *Csp* = *Clostridium sp.*, *Bvu* = *Bacteroides vulgatus*, *Cce* = *Clostridium celatum*.

|  | Sex | Hatch year | Bird # | Sample Date | 16S reads | Metagenomic reads | % reads recognized | Dominant species |
|---|---|---|---|---|---|---|---|---|
| Cloacal | M | 2013 | 697 | 17-Dec-13 | 39,234 | 11,126,438 | 15.8 | Cje, Pac |
|  | F | 2013 | 707 | 18-Dec-13 | 43,537 | 12,056,810 | 15.5 | Cje,Sfl |
|  | M | 2013 | 706 | 8-Jan-14 | 0 | 13,453,902 | 3.8 | Sfl, Cje |
|  | M | 2013 | 716 | 6-Jan-14 | 23,651 | 14,265,278 | 8.4 | Sfl, Cpe |
|  | F | 2013 | 721 | 22-Jan-14 | 28,771 | 15,460,744 | 4.3 | Sfl, Fva |
|  | M | 2013 | 685 | 14-Dec-13 | 55,024 | 13,786,200 | 17.2 | Bov, Cpe |
|  | M | 2013 | 696 | 17-Dec-13 | 15,659 | 13,051,190 | 28.2 | Ckr, Eco |
|  | F | 2013 | 698 | 16-Dec-13 | 34,056 | 8,506,852 | 12.7 | Ckr, Sps |
|  | M | 2013 | 700 | 16-Dec-13 | 19,534 | 8,994,382 | 24 | Ckr, Pav |
|  | M | 2013 | 701 | 18-Dec-13 | 25,682 | 15,598,718 | 16.7 | Dac, Ckr |
|  | M | 2013 | 704 | 20-Dec-13 | 29,782 | 13,051,190 | 18.3 | Dac, Eco |
|  | M | 2013 | 699 | 27-Dec-13 | 19,364 | 10,933,336 | 7.3 | Cul, Dac |
|  | M | 1991 | 60 | 14-Dec-13 | 25,713 | 11,785,836 | 13 | Dac, Bsu |
|  | M | 2013 | 711 | 21-Dec-13 | 8,207 | 11,974,718 | 37.5 | Dac, Esp |
|  | M | 2013 | 703 | 27-Dec-13 | 21,495 | 15,304,924 | 18.5 | Cac, Sma |
|  | F | 1997 | 166 | 14-Dec-13 | 7,474 | 13,353,706 | 12.6 | Dac, Cje |
|  | M | 2010 | 580 | 15-Dec-13 | 2,772 | 12,457,476 | 57.7 | Dac, Sma |
|  | F | 1991 | 69 | 15-Dec-13 | 2,149 | 14,452,108 | 51.6 | Dac, Sma |
|  | M | 2013 | 718 | 9-Jan-14 | 3,685 | 12,140,744 | 14.5 | Dac, Sma |
|  | F | 1991 | 59 | 12-Dec-13 | 2,104 | 11,789,408 | 45.4 | Dac, Sma |
| Fecal | M | 2013 | 685 | 14-Dec-13 | 47,014 | 16,659,098 | 48.6 | Cpe, Ljo |
|  | F | 2013 | 698 | 16-Dec-13 | 29,588 | 15,149,972 | 32.8 | Cpe, Fmo |
|  | M | 2013 | 716 | 6-Jan-14 | 41,898 | 16,279,536 | 39.1 | Cpe, Fva |
|  | M | 2003 | 309 | 1-Oct-13 | 42,610 | 15,212,904 | 45.2 | Cpe, Pan |
|  | M | 2013 | 696 | 17-Dec-13 | 49,105 | 16,654,564 | 43.5 | Csp, Fva |
|  | M | 2013 | 704 | 20-Dec-13 | 32,171 | 15,526,658 | 38.8 | Cpe, Fva |
|  | M | 2010 | 580 | 15-Dec-13 | 37,652 | 15,745,052 | 38.2 | Fva, Cpe |
|  | M | 2013 | 699 | 27-Dec-13 | 47,374 | 15,381,106 | 45 | Cpe, Fva |
|  | M | 2013 | 309 | 7-Oct-13 | 0 | 14,752,242 | 20.8 | Fva, Cce |
|  | M | 2013 | 701 | 18-Dec-13 | 32,323 | 15,346,160 | 31.2 | Bvu, Sfl |

bacteria are present representing the genera *Corynebacterium*, *Shigella*, *Fusobacterium*, and *Staphylococcus*. The mature cloacal samples were dominated by a single organism, *Delftia acidovorans*. *Clostridium* and *Fusobacterium* were common genera in the fecal samples, although one sample was composed predominantly of *Bacteroidetes* and *Shigella*.

When analyzing data at the node level (data from Fig 2), species richness significantly differed among the three groups (ANOVA: $F_{2,27} = 33.95$, $P < 0.001$). Again, all three groups significantly differed (Tukey post-hoc test: all p-values < 0.05) and the cloacal immature samples had the highest number of species. Cloacal mature had the fewest number of species. The three groups significantly differed in terms of Simpson's diversity (ANOVA: $F_{2,27} = 55.09$, $P < 0.001$). Cloacal mature had lower microbial diversity than cloacal immature (Tukey post-hoc test: $P < 0.001$) and fecal mature (Tukey post-hoc test: $P < 0.001$) samples.

**Phylogenetic rollup of condor microbiome composition, computed with Sequedex / SEED**

**Fig 1. A stacked bar chart of the relative abundance of the bacterial composition of each of the samples is shown for selected phyla, indicated to the right of the Fig.** Predominant species comprising each phylum are indicated on the bar chart. Bacterial prevalence is estimated for each sample by Sequedex classification of the shotgun metagenomics data, and species were identified with both 16S sequence data (296 nucleotide per read) and selected RNA polymerase reads identified by Sequedex as being phylogenetically informative. Forward and reverse reads were analyzed separately and were closely related. The separation into four groups was made on the basis of sample type, (cloacal vs. fecal) and phylogenetic and functional profiles (cloacal: immature vs. mature). Complete results are provided in S1a File.

Cloacal immature and fecal mature did not significantly differ from each other (Tukey post-hoc test: P = 0.81).

Although not based on an alignment, Sequedex analysis of shotgun metagenomic reads allows the comparison of between-sample, or beta, diversity (Fig 2). In practice, highly populated reads typically occur in related groups, discussed above and shown in S1 Fig, so it is possible to assign these groups of nodes to particular species present in the family by sampling a few dozen phylogenetic marker reads (eg. RNA polymerase) from each node and examining the returned hits with BLASTN from NCBI's nonredundant database. These species names are provided to the left of the heat map in Fig 2. Complete results across all phylogenetic categories and samples is provided as a spreadsheet in S1 File.

While the analysis shown in Fig 2 does not differentiate subtle phylogenetic differences between samples, the general similarity of intensities across related nodes provides an indication of the similarity of community composition across samples. This analysis is related to OTU analysis based on alignments of 16S reads, as can be seen in a heat map of the occupancy of the most prevalent OTUs (S2 Fig). As with Fig 2, approximately 90% of the assignable bacterial reads are covered by the analysis in S2 Fig, so we expect similar organisms to be found. Because QIIME places OTUs on a taxonomy instead of a tree, taxonomic assignments are often restricted to the genus or family level, even at an 80% confidence level. Unlike with the RNA polyerase reads used in Fig 2, the results were not typically specific enough to make species assignments. To resolve ambiguous cases, we used species assignments that matched Fig 2 if occurring, provided a genus level assignment if several species matched equally well, or simply picked a well-known species to list if the other matches were named only to the genus level.

Another useful OTU analysis enabled by 16S data are the rarefaction curves, shown in Fig 3. We plotted the number of 16S sequences that must be sampled to observe a given number of species. We used the R package 'iNEXT' [29] for the rarefaction and pooled individual birds for the three groups. The species richness between the three groups significantly differed based on the 95% confidence intervals of the three groups not overlapping (Fig 3). This analysis

Species identified from metagenomic reads

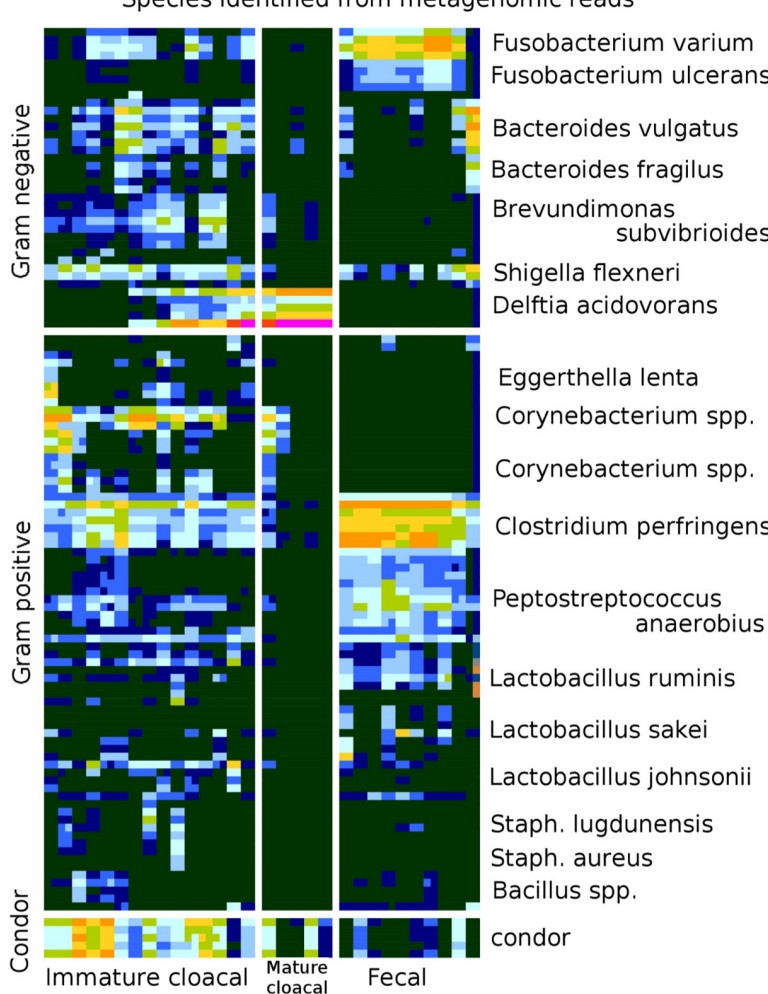

**Fig 2. Sample-to-sample variability of phylogenetic composition across microbiome samples is computed by comparing the normalized number of reads recruited to specific nodes on the phylogenetic tree with Sequedex.** The number of reads in each sample that were assigned to the most-populated nodes of the reference phylogenetic tree (S1 Fig) are shown as rows in this heat map. These rows represent nineteen of the most common species observed, and the color indicates relative abundance, with dark blue indicating zero, and red more than half of the sample. These most-populated nodes account for more than 90% of the bacteria present in the sample without needing to differentiate the case of dominant species from diverse communities in particular areas of the phylogeny. Species names were determined using BLASTN with phylogenetically discriminating reads from the rpoB and rpoC polymerase reads. Complete results are provided in S1b File.

distinctly shows the higher diversity of the cloacal samples from immature birds, compared to cloacal samples from mature birds or fecal samples.

We further analyzed the data rolled up into phyla (from Fig 1) to compare the community composition among the three groups (immature cloacal, mature cloacal, and mature fecal). The three groups significantly differed overall in microbial composition based on phylogenetics (PERMANOVA: $R^2 = 0.49$, $P < 0.001$; Fig 4a). Each group was significantly different than the other two groups after analyzing the pairwise comparisons (immature cloacal vs. mature cloacal: $R^2 = 0.30$, $P < 0.001$; mature cloacal vs. mature fecal: $R^2 = 0.56$, $P = 0.002$; immature cloacal vs. mature fecal: $R^2 = 0.41$, $P < 0.001$), suggesting a unique microbiome for each group.

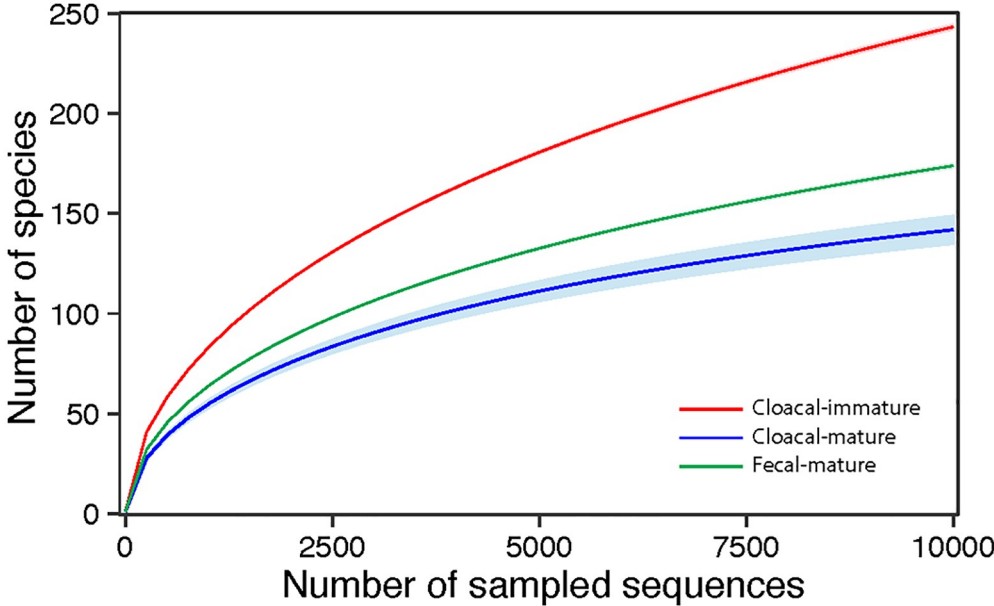

**Fig 3. 16S rarefaction curves.** Within-sample richness, estimated by OTU analysis of 16S sequences with QIIME [32]. The rarefaction curves are computed by sampling the number of 16S sequences indicated on the x-axis and counting the number of distinct observed species, plotted on the y-axis. The shaded areas around the lines represent the 95% confidence intervals. None of the confidence intervals overlap, suggesting that each group significantly differs from the other two even when only 10,000 sequences are considered.

Anticipating the functional composition results described below, Fig 4b provides an analogous analysis across the 962 functional categories provided by Sequedex. The functional gene categories also significantly differed between the three groups (PERMANOVA: $R^2 = 0.71$, $P < 0.001$; Fig 4b). Each pairwise comparison was statistically significant (immature cloacal vs. mature cloacal: $R^2 = 0.64$, $P < 0.001$; mature cloacal vs. mature fecal: $R^2 = 0.81$, $P < 0.001$; immature cloacal vs. mature fecal: $R^2 = 0.46$, $P < 0.001$). Both the phylogenetic and functional Sequedex output files used in these analyses are provided in spreadsheet form in S1 File.

None of the analyses described above provide insight into fine-scale phylogenetic structures within communities. The shotgun metagenomic reads are not in an alignment, while the 16S reads lack the length and resolution to provide such information. For the mature cloacal samples, however, greater than 80% of microbial population appears to be *Delftia spp.*, and it is possible to create alignments of the RNA polymerase gene, a particularly informative phylogenetic marker gene. The branch length for each of the four genes assembled from each sample differs, indicating the presence of either sequencing errors or genetic diversity in *Delftia* strains present within the sample (S3 Fig). The results also indicate that individual birds can be discriminated from each other, as the reads for each bird are grouped together in a tree, so it is possible to distinguish birds by where their assembled microbiome reads fall in the tree.

## Functional composition of microbiomes

As mentioned above, shotgun metagenomics reads analyzed by Sequedex provide an ability to analyze the functional composition of the metagenome in a manner independent of the phylogenetic analysis. Fig 4b illustrates that both analyses provide a similar ability to discriminate among samples.

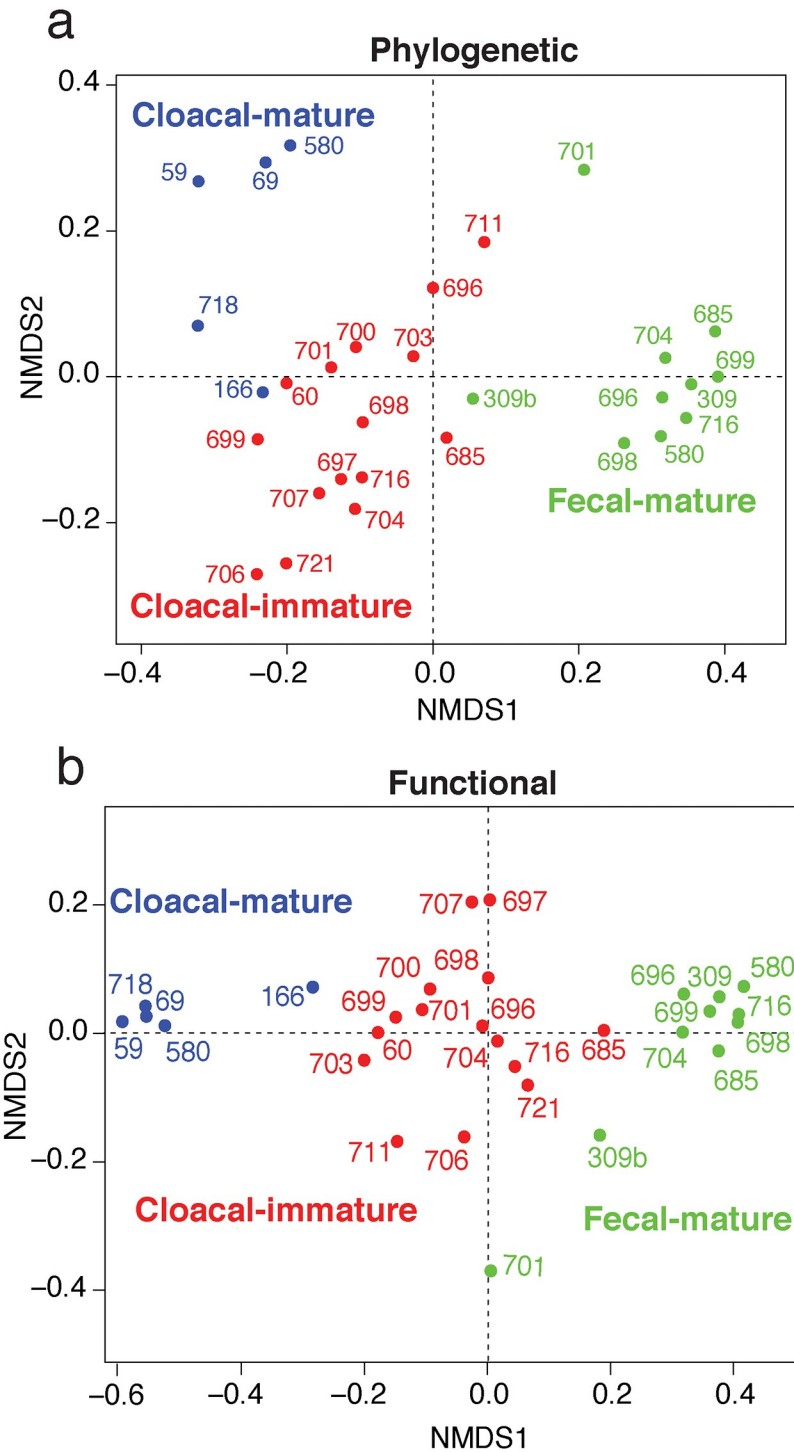

**Fig 4. Plot of non-metric multidimensional scaling (NMDS) of composition of the three groups.** The three groups significantly differ in their microbial composition for both a. phylogenetic classification, and b. functional gene classification.

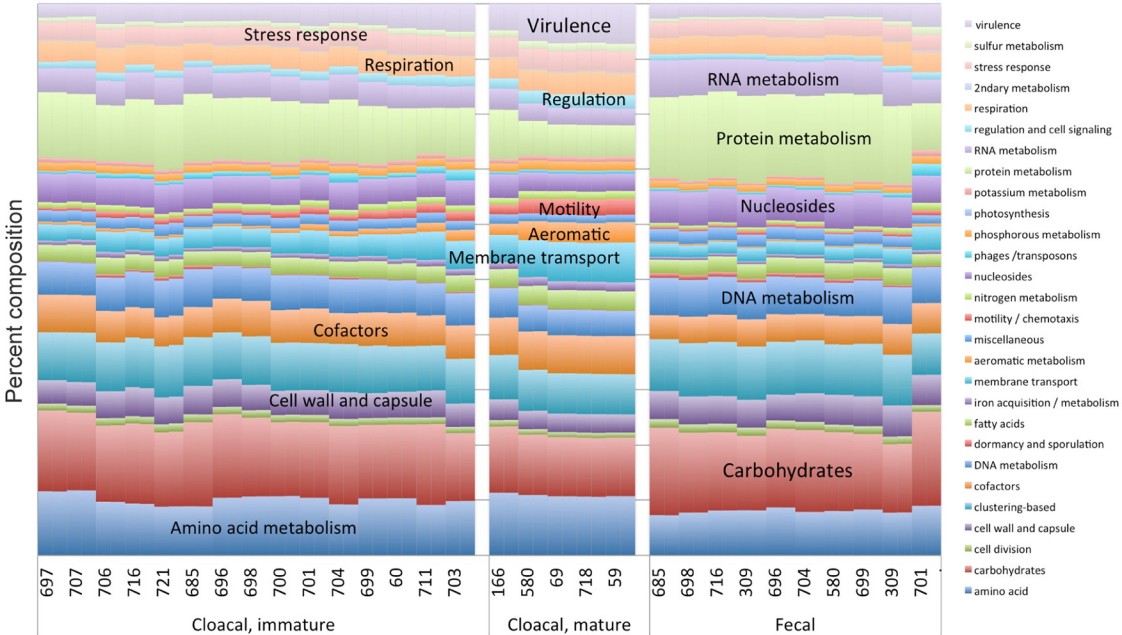

**Fig 5. A.** Functional rollup of read-based classification of the DNA or RNA in each sample. Classification was performed on each read with Sequedex and the SEED subsystems. The legend at right provides the color code for all 28 functional categories, while labels on the plot indicate highly populated categories as well as those that change significantly between types of samples. Complete results for the functional rollups are provided in Supporting Information S1d File. **B.** Subsystems associated with fecal metabolism. Selected subsystems distinctive of the fecal microbiome are plotted, with the y-axis measuring the abundance of functionally identified reads associated with the particular subsystem, for each of the samples. **C.** Subsystems associated with cloacal metabolism. Selected subsystems distinctive of the cloacal microbiome are plotted, with the y-axis measuring the abundance of functionally identified reads associated with the particular subsystem, for each of the samples.

In doing the Sequedex analysis, we chose to use SEED subsystems to characterize the functional complement of genes present in the samples in part because they provide a hierarchical classification of protein function. Although high-level categories are generally within a factor two of representation in the genomic DNA of the community, several differences are evident by inspection, such as the increased protein metabolism in the fecal samples and the increased virulence and transport-related genes in the cloacal samples.

At the highest resolution of the SEED classification, several notable subsystems are enriched in the different sample types. Metabolism and cell wall subsystems increase notably in fecal samples, while virulence, motility, membrane transport, and sulfur metabolism increase in the mature cloacal samples (Fig 5b). Nitrogen metabolism is elevated in immature cloacal samples, while phages are greatly elevated in the *Bacteroidetes*-dominated fecal sample at right, when compared to the Firmicutes-dominated fecal samples (Fig 5c).

A birds-eye (so to speak) view of how all 962 SEED subsystems are represented in the thirty samples is shown in Fig 6. While no distinctly 'immature cloacal' functional subsystems can be identified, subsystems elevated in fecal samples are primarily associated with anaerobic metabolism and growth, while subsystems elevated in mature cloacal samples are primarily associated with nitrogen metabolism and surviving in the presence of toxins. Highly abundant SEED subsystems found in the California condors include Acetyl-CoA fermentation to butyrate, methionine degradation, glycolysis and gluconeogenesis, and several different translation elongation factor subsystems. The complete breakdown of functional categories across all thirty samples is provided in spreadsheet form in S1 File.

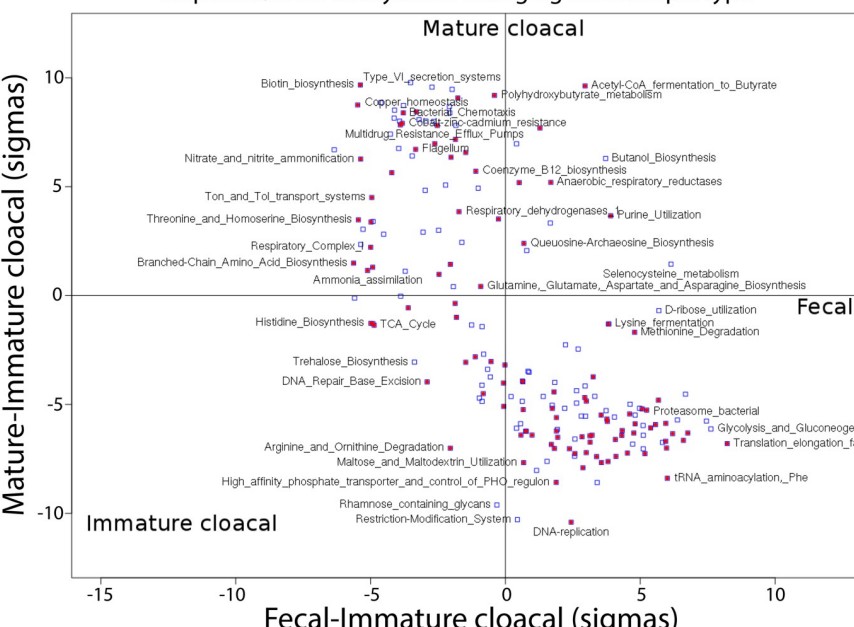

**Fig 6. SEED subsystem with significant changes between sample types.** Z-scores for the difference between fecal and immature cloacal samples on the x-axis are plotted against z-scores for the difference between mature and immature cloacal samples on the y-axis for the two thirds of the most-occupied functional categories. For clarity, only the most populated 2/3 of the SEED subsystems are shown, with the filled magenta squares representing the most populated 1/3 of subsystems and the clear squares are the middle group. Using the clusters of samples defined with the PCA analysis, specific subsystems are identified according to their relative representation in fecal vs. immature cloacal samples (x-axis) and mature vs. immature cloacal samples (y-axis) in units of difference divided by the standard deviation estimated across samples. These categories provide greater resolution to the changes identified in the rollups, and complete results are provided in S1d File.

## Discussion

In this study we sampled captive California condors to determine individual variation in the gut microbiome of a captive scavenging species. Previous studies have found that the bacterial repertoire in wildlife depends mainly on diet and host phylogeny [30, 31] as well as whether avian samples are fecal or cloacal [32]. At this point in time, wild condors are not available for comparison. However, we predict that wild condors will have a more diverse microbiome than captive individuals. We found that immature condors had much more diverse microbiomes than adult birds. This is unlike poultry and Chinstrap Penguins (*Pygoscelis antarctica*), in which microbiome diversity increases with age [23, 24]. However, it does follow the results of Teyssier et al. [6] who found cloacal microbiota of Great Tit nestlings decreased in diversity and increased relative abundance of Firmicutes between 8 and 15 days of age. While the process for the differences of the microbial community is unclear, it will be important to continue to compare species for the development of microbial communities with age of individuals.

The two methods (16S and metagenomic) largely agree on the ~ 20 species that comprise approximately 89% of the assignable genomic reads found in the condor samples. However, the two methods disagree significantly on the relative abundance of each species in the bacterial community. One of the largest discrepancies is in the prevalence of *Delftia* in the mature cloacal samples, which the shotgun metagenomics identifies as 80% of mature cloacal samples, but which barely registers in the 16S data (compare Figs 1 and 2 to S2 Fig). Given our ability to assemble multiple copies of the *Delftia* RNA polymerase gene from these samples, it appears

far more likely that the shotgun metagenomic estimate is the more reliable, presumably due to the lack of primer-specific amplification biases. This behavior is consistent with other studies comparing species abundance between targeted 16S sequencing and shotgun metagenomics approaches [33]. The complete distribution of OTUs across the condor samples is provided in S1c File, and a FASTA format file with the representative 16S sequences for each OTU in S1c is provided as S2 File.

Grond et al. [12] provides a recent review of studies on avian gut microbiota. In this work, they note the broad agreement across studies of the presence of firmicutes, bacteroidetes, proteobacteria, and actinobacteria, consistent with our findings. In view of the significant discrepancies of 16S profiles compared to shotgun metagenomic, it appears that caution is necessary when making more detailed comparisons across host species and environmental conditions until more studies characterize the repeatability of various sampling methods across samples. Additionally, the increased phylogenetic resolution afforded by 150 base pair reads of phylogenetic marker genes over that of 16S reads would improve comparison across papers from phylum- or genus-level to species-level comparisons.

Roggenbuck *et al.* [14] describe several near-neighbors to pathogens in vultures: Clostridia and Fusobacteria dominated vultures' fecal and hindgut samples. Some members of these bacterial classes are pathogenic to other vertebrates, and may also be pathogenic to vultures. If so, it is likely that vultures tolerate toxins from pathogenic bacteria obtained from their carrion diet because they benefit from breakdown of carrion by those same bacteria. We found the classes Clostridia and Fusobacteria to dominate fecal samples from California condors, highlighting the importance of the scavenger lifestyle in captivity as well. In a recent microbiome study of the microbes in Large-Billed or Jungle Crow (*Corvus macrorhynchos*), it was found that Jungle Crows not only acquire pathogenic microbes, but they return the microbes to the environment via feces [34]. If this pattern is similar for wild California condors, they may play an important role in spreading pathogens in the environment. Thus, this species, as well as vultures, could be used as sentinel species to understand the pathogens present in certain ecosystems.

## Microbial composition

The California condors we sampled in this study showed considerable diversity but share a common core of microbes. This could be due to the captive birds being fed the same food but is more likely related to the birds' shared physiology. Roggenbuck et al. [14] found captive vultures from a zoo house together shared gut microbes with wild vultures while differing from non-vulture birds sharing their captivity and diet. Like vultures, condors are detritivores, and the condors used in this study were all fed the same raw diet. The observed variation in microbiomes could be due to varying preferences for lean or fatty meat types or could be caused by a condor's social status leading to differences in diet among animals in the same enclosure.

In addition, as we were only able to sample DNA for this study the diversity found may be under-representative of that actually present. Much greater differences are expected for mRNA samples, in which only genes being utilized in the particular environment sampled will be transcribed.

*Fusobacteria* are obligate anaerobes that ferment amino acids, peptides, and carbohydrates that are often found in the mucus layers of humans and animals [35]. The phylogeny of the genus *Fusobacteria* is shown in S1 Fig, and greater than 99% nucleotide identity, was observed for RNA polymerase reads to all of the following species: *Fusobacterium varium*, which is a gastrointestinal species; *F. ulcerans*, which has been isolated from tropical ulcers [35]; and *F. mortiferum*, which is an anaerobic human pathogen that ferments carbohydrates [36].

The surface of the cloaca is a mucus membrane in which commensal or pathogenic bacteria may be found [37]. It provides both the first line of defense and an area where bacteria such as *Lactobacillus spp.* can attach to the host. *Delftia* spp. and *Comamonas* spp. are microorganisms that often comprise the bulk of bacteria in biofilms associated with wastewater treatment and river wetlands [38, 39]. The organisms in the majority of the condor samples that are highlighted here are associated with robust biofilm formation, antibiotic resistance, and eye infections [40–42].

As noted previously, many of the organisms found in meat-eating birds are closely related to bacteria isolated from humans and birds [14]. The bulk of bacteria in the fecal microbiomes are in the classes Clostridia and Fusobacteriia, which follows the results of the study by Videvall et al. [32], who compared fecal and cloacal microbiomes in ostriches (*Stuthio camelus*). Specifically, for Clostridia, the predominant species of most of the fecal samples in this study, with reads having greater than 98% nucleotide identity to RNA polymerases in reference strains, was *Clostridium perfringens*. *Clostridium perfringens* is a poultry pathogen that causes necrotic enteritis if predisposing factors are present [43].

From our functional analyses (see Fig 4), it is clear the functional categories are distinct between fecal and cloacal samples. While the classes Clostridia and Fusobacteriia make up the bulk of the bacteria in the fecal microbiomes, the genus *Delftia* makes up the bulk of the bacteria in the mature cloacal microbiomes. This variance we see between the fecal and mature cloacal microbiomes is also supported by Videvall et al. [32], who concludes that it is best to use fecal samples as a representative of the colon in birds and that fecal and cloacal samples only represent the lower digestive tract of the colon. Our use of both fecal and cloacal samples provides complementary information on the California condor microbiome.

Some organisms identified in California condors are quite close to reference genomes despite being novel species. Examples of these species follow. Ninety-three percent amino acid identity was observed to be *Propionibacterium acnes*, a non-spore-forming, Gram-positive, opportunistic pathogen that is part of normal human microflora and found in the skin, oral cavity, GI, and genito-urinary tracts [44] *Bacteroides fragilis* reads were found with greater than 99% nucleotide identity to the RNA polymerase of reference genomes in both fecal and cloacal samples. *Bacteroides fragilis* is an obligate anaerobe that is a normal component of human gut flora, but is a common cause of abscesses and bacteremia [45–47]. Ninety-seven percent amino acid identity of RNA polymerase reads was observed to *Eggerthella* YY7918, a species that is often part of the human intestinal flora and has been identified as a cause of severe [48]. This species was found at a much higher abundance in the cloacal samples than the fecal ones. *Shigella flexneri* is a non-motile facultative anaerobic gram-negative pathogen that causes dysentery in humans [49, 50] and is increasingly antibiotic resistant [51].

Other organisms identified in California condors are not very close to reference genomes. For example, *Shingella flexneri* was found at levels of 5–10% in the condor fecal and immature cloacal samples. Consistent with Fig 3, the fecal and mature cloacal samples have considerably less within-sample diversity than the immature cloacal samples. While this method is free of some biases, the presence of sequencing errors in 16S data can lead to significant over-estimation of the number of distinct sequences in a sample. The species richness is lower than that observed by comparable studies in some bird species (see [9]), but is consistent with that observed in vultures [14].

## Functional composition of microbes

Functional profiles separate the microbial communities in a manner consistent with phylogenetic profiles, and with a similar dispersion among samples (Fig 4). We also showed our ability

to identify the determinants of functional classification, both at the coarse-grained level (fecal samples have more protein and carbohydrate metabolism genes, while cloacal samples show more transport and motility genes) and at the fine-grained level (fecal samples show more replication and methionine degradation, while cloacal samples show more nitrate metabolism and anti-toxin systems).

Genome inventories of bacteria are often quite different from their near-neighbors, or not-so-near-neighbors, which happen to have been sequenced and deposited in Genbank. Wu et al. 2015 [39] describe the importance of nitrogen metabolism to the biofilm lifestyle of *Comomonas*, and it is natural to wonder if the *Delftia* species we see as a major component of the mature cloacal samples contains nitrogen reduction capabilities. The two copies of the alpha, beta, and gamma subunits of nitrate reductase provided in S3 File indicate that this phenotype is present in the *Delftia* species inhabiting the condors we evaluated.

Following the example of Roggenbuck, *et al.* [41], we looked for *Clostridium spp.* toxins in our samples, and do present partial sequences of the alpha toxin in S2 File. While more investigation is needed to understand why we did not see the entire gene, we were able to recover significant quantities of colicin sequences, and they are recorded with their sequences. While such detailed analysis is quite laborious, it appears likely to be necessary if functional characterization of microbiomes is to reach a useful level of detail, as described by Grond et al. [12].

Determining the phylogenetic composition of a natural microbial community from the sequence of fragments of DNA is complex both because of the great variability in the mapping between sequence similarity and phylogenetic similarity for different genomic fragments, and because natural microbial communities often contain organisms that are quite distinct from anything that has been cultured, sequenced, and placed in a reference database [28] (see S3 Fig for a phylogenetic tree of the *Delftia* RNA polymerase genes recovered from cloacal samples compared to those from nearby reference genomes used in the construction of Sequedex's database). We chose to compare the samples with conserved signature peptides for which the level of phylogenetics has been characterized, in order to address both of these issues. These concepts are discussed at length in the Sequedex paper [28]. Sequedex uses 300,000,000 amino acid signatures of length 10 to assign genomic fragments individually to the 2550 nodes on its one-per-species phylogenetic tree of life. In each case, further investigation can ascertain the phylogenetic composition with more accuracy.

## Conclusion

We characterized gut microbiota from healthy captive California condors to facilitate the design of future studies and to demonstrate novel methods that can be applied to other systems. Useful microbiome signatures were identified by comparing samples within and between individual condors, using both standard and novel methods. In sampling captive and wild animals, one must make a number of choices about sample type: sequencing DNA or RNA, the necessary depth of sequencing, sequencing 16S reads or at random, the types of birds (e.g. sick or healthy, captive or wild, young or old) to be differentiated, if sampling will be done in a time series analysis or at isolated points in time, and the number of locations to be sampled from.

In this work, we were able to identify distinct species of microbes in cloacal and fecal samples, as well as interpretable differences in the functional complement of genes in the two groups of samples. One of the most interesting findings from this study is the 40% higher diversity in the immature birds compared to adult condors.

The Sequedex software package has enabled rapid analysis of such data sets in a few hours on a laptop computer. It provides both readily interpretable output files profiling both

phylogeny and function without complex, non-linear processes such as assembly, and the annotation of raw reads to enable follow-up confirmatory analysis. When combined with appropriate sampling of, for example, a sentinel species at the top of the food web, the potential to gather information rich data across the globe will serve an acute need in monitoring climate change impacts on ecosystems in order to guide mitigation efforts. Furthermore, if successful, this work opens the possibility of using microbiome profiles and disease surveillance of other sentinel species to provide urgently needed specific signatures of ecosystem health [52].

## Materials and methods

### Ethics statement

The methods used in this study were approved by the Los Alamos National Laboratory (LANL) Institutional Animal Care and Use Committee and did not affect the health and welfare of the animals. All State and Federal wildlife and captive breeding permits are held by the Peregrine Fund (Boise, ID). All laboratory procedures for biological samples were approved by the LANL Biosafety Committee.

### Sample collection

Our study samples are from 22 captive adults and immature (hatch year) birds from the rearing facility of the Peregrine Fund in Boise, Idaho. Since 1996, the Peregrine Fund has been breeding and raising California condors in captivity for release to the wild. The sampled condors lived in captivity since hatching, were fed a controlled diet of mostly rodents, rabbits, and commercial bird of prey diet, and were housed in a single building. All of the birds were housed together and are the same diet daily. While birds may be housed separately between cages, due to the open structure of the facility and movement of the birds, they can be considered being raised together without isolation from each other in regards to exposure to microbes and eating the same diet.

Fecal samples were collected opportunistically from October 2013 to early January 2014, from both adult and immature condors within 30 minutes of the time of defecation. Cloacal samples were collected during routine health checks by the facility personnel and veterinarians. Both fecal (n = 10) and cloacal (n = 20) samples were placed in sterile plastic specimen tubes when collected and stored in a -20˚C freezer within 10–30 minutes of being acquired. Cloacal samples were saved on dry flocked swabs (HydraFlock, Puritan Guilford, ME) in the sterile plastic specimen collection tubes with no reagent. Sampling strategy was designed primarily by sequencing costs and number of birds available. Ten condor fecal samples (9 immature birds and 1 adult) and 20 cloacal samples (15 immature birds and 5 adults) were taken for analysis from different individuals for a total of 30 samples sequenced.

### DNA isolation

Immediately prior to analysis, frozen fecal samples were diluted with water and vortexed. Cloacal swabs were resuspended in water by vortexing. Lysis buffer from the DNA Fungal/Bacterial DNA isolation kit (Zymo Research, Irvine, CA) was added to the samples, and lysis was performed for 40 seconds in ZR bead bashing tubes (Zymo Research, Irvine, CA) using the Fastprep 24 instrument (MPBio). Cloacal DNA was purified using the same kit (Quick DNA Fungal/Bacterial Miniprep Kit, Zymo Cat. #D6005), as per the manufacturer's protocol. DNA was quantified using Qubit instruments and ranged from below the level of detection to 60 ng/ul. The integrity of DNA was evaluated using 1% agarose gels with Lambda DNA/HindIII Marker (Thermo Fisher Scientific, Cat. #SM0102).

## Library preparation and sequencing

For shotgun metagenomics samples, Illumina libraries were prepared using NEBNext Ultra DNA Library Preparation Kit (New England Biolabs, Cat. #E7370S) per the manufacturer's protocol. Due to the varying amounts of DNA extracted, the input amount of DNA ranged from 100 ng to less than 1 ng. DNA was fragmented using a Covaris E220, and the ends were made blunt and adapters and indexes were added onto the ends of the fragments prior to PCR enrichment. For samples with inputs below 10 ng of DNA, the number of PCR cycles was increased from the manufacturer's protocol of 12 to 15. The Illumina metagenomic libraries were cleaned up using AMPure XP beads (Beckman Coulter, Cat. #A63881) before being eluted in DNA Elution Buffer (Zymo Research, Cat. #D3004-4-10). The concentration of the libraries was obtained using the Qubit dsDNA HS Assay (ThermoFisher Scientific, Cat. #Q32854). The average size of the library was determined by the Agilent High Sensitivity DNA Kit (Agilent, Cat. #5067–4626). An accurate library quantification was determined using the Library Quantification Kit–Illumina/Universal Kit (KAPA Biosystems, Cat. #KK4824). The libraries were sequenced on the Illumina NextSeq generating paired-end 151 bp reads.

For 16S sequencing, degenerate primers that amplify the V4 region of bacterial rDNA genes were used (515–806 pair). Due to the varying amounts of DNA extracted, the DNA input ranged from 12.5 to less than 1 ng. The first round of PCR amplified the V4 region and added unique tags to the amplicon, using a denaturation temperature of 95˚C for 3 minutes, 25 cycles at 95˚C for 30 seconds, 55˚C for 30 seconds and 72˚C for 30 seconds, followed by an extension of 72˚C for 5 minutes before holding at 4˚C. The second round of PCR added Illumina specific sequencing adapter sequences and used a denaturation temperature of 95˚C for 3 minutes, 8 cycles of 95˚C for 30 seconds, 55˚C for 30 seconds, and 72˚C for 30 seconds, followed by an extension of 72˚C for 5 minutes before holding at 4˚C. The amplicons were cleaned up using AMPure XP beads (Beckman Coulter, Cat. #A63881). A no-template control was processed but did not show a band in the V4 amplicon region and was not further processed. Unique tags allowed for multiple amplicons to be pooled together. The concentration of the amplicon pool was obtained using the Qubit dsDNA HS Assay (ThermoFisher Scientific, Cat. #Q32854). The average size of the library was determined by the Agilent High Sensitivity DNA Kit (Agilent, Cat. #5067–4626). An accurate library quantification was determined using the Library Quantification Kit–Illumina/Universal Kit (KAPA Biosystems, Cat. #KK4824). The amplicon pool was sequenced on the Illumina MiSeq generating paired end 301 bp reads.

## Genomic analysis

Shotgun metagenomes were analyzed for phylogenetic and functional content according to the 2550-species tree of life, and 963 SEED functional categories utilized with the Sequedex package [28] (http://sequedex.lanl.gov/). Sequedex is a signature-peptide based analysis that assigns both phylogeny and function independently to each read based on its database of 300 million signature peptides. It avoids many of the common pitfalls of assembly-based, blast-based, or gene-based analysis methods and is extensively described in Berendzen et al. 2012.

While techniques exist to attempt to identify individual component taxa in a complex sample [53], we chose in our case to simply recruit reads from these samples to amino acid reference sequences for the RNA polymerase genes, using the reading frame for each read identified by Sequedex. In this case we can assemble a phylogenetic marker gene long enough for detailed phylogenetic analysis, and provide some indication of whether different birds have distinct species present. A direct evaluation of the functional profile of the microbiome was obtained from the read-based functional assignment to the hierarchical, SEED subsystem

classification scheme, using Sequedex as well [28]. As with phylogeny, binning of reads into generally-defined functional categories can be misleading. The by-read annotation provide by Sequedex allowed us to assay for not just total representation of particular SEED subsystem, but also greatly facilitated our ability to retrieve specific genes from shotgun metagenomic samples. We demonstrated this ability on five representative genes: the phylogenetic marker rpoB, a metabolic niche gene, nitrate reductase, a beta lactamase, and a *Clostridium* toxin. The S3 File provides the sequences of example genes.

In phylogenetic signature methods, high-significance patterns that are shared across taxa are identified with positions on a pre-calculated tree via a Least Common Ancestor algorithm. In principle, it matters little whether these signatures are genetic or anatomical. However, for genetic signatures, some patterns appear in multiple species for reasons other than phylogeny (e.g., triplet repeats of DNA that may be artifacts of the replication mechanisms, or horizontally transferred genes appear because of common ecological niches). In our analysis, those candidate signatures that are part of a repetitive region of sequence were discarded, and the classifying power of the remaining set of signatures were characterized. Good signature sets have both high selectivity and high sensitivity. We characterized the signature set we used as having similar sensitivity to BLASTX, while having more than an order of magnitude higher selectivity when used for phylogenetic classification of short reads such as those used in this experiment [28].

The functional profiles from Sequedex provided support and additional insight to the niche determinants derived from the microbiology of the particular species of organisms found in each sample. Both phylogenetic assignments and genome inventory analyses were performed on the samples, allowing the bacteria identified to be separated into three probable groups: microbes of importance to digestion, those associated with mucosal environments, and potential pathogens of interest.

In this work, we chose to identify specific reads from a phylogenetic marker gene, the RNA polymerase, for further examination using BLASTN against the non-redundant database at NCBI. Sequence reads corresponding to a phylogenetic marker gene, RNA polymerase, were identified from each of these nodes and assigned a taxonomy using NCBI to blast a nucleotide fragment (blastn) against the nucleotide (nt) database. After examination of about a dozen reads in this way, species can be identified for each set of nodes.

By examining the SEED subsystems identified in S1e File, as well as the number of reads assigned to particular Pfam families in S1f File, we were able to find samples and genes likely to have high enough prevalence to be assembled and compared to the literature. We provide two examples of beta lactamases in S2 File, where a reference gene was used as a template, and translated peptide sequences were matched to it by identifying exact 10-mer matches to the reference. Two examples were recruited for each reference, in order to provide some idea of the variability of the resulting sequence that is recruited by this method.

The analysis method we use here, Sequedex, is an example of a signature-based analysis method [28]. Signature methods have a distinguished history across a range of scientific disciplines as a way of identifying genes of value in situations where application of fundamental theory may be complex. Hidden Markov Models of amino-acid sequences are an example of signature methods, in contrast with BLAST, which is usually used as a profiling method.

## Statistical analysis

We compared measures of community structure between the three groups (age and sample type), including species richness, species diversity, and community composition. For species diversity, we used Simpson's diversity index. We tested for differences in richness and diversity

between these groups using one-way ANOVAs followed by Tukey's post-hoc test to test for pairwise differences between groups. Assumptions of normality and equal variances for parametric tests were met.

Community composition among the three groups (age and sample type) were compared using non-metric multidimensional scaling (NMDS) using the metaMDS function in the vegan package (version 2.4–4; [54]) in R (version 3.4.1; [55]). Two separate NMDS ordinations were done; one based on phylogenetics and the other based on functional gene categories. To test for overall differences among groups, Bray-Curtis distances were calculated and then examined using Permutational Multivariate Analysis of Variance (PERMANOVA). The adonis function (vegan) was used with 1000 permutations. To test for differences between each group, pairwise PERMANOVAs were run, each with 1000 permutations, and resulting p-values were compared to a Bonferroni corrected alpha of 0.017. A better understanding of the diversity of organisms present in the samples can be obtained through phylogenetic analysis. We compared separate copies of RNA polymerase genes assembled from either the forward or reverse reads in each sample. The assembled genes are closest to the two *Delftia* reference genomes, with one of the two being *Delftia acidovorans* and the other reference not assigned a species name. The observed phylogenetic separation of the genes from the condor microbiome appear to match the reference Delftia at the genus level but not at the species level. The four genes assembled from each sample cluster together on the tree, indicating the organisms from each bird are genetically distinct from each other.

The 16S data analysis was performed with QIIME version 1.8 [56] with representative 16S sequences are provided in S2 File, and were selected using the QIIME utility, single_rarefaction.py, and a value for a sequence similarity parameter of 0.94. Annotations of 16S sequences were made by comparing to the 97_otus.fasta database.

## Supporting information

**S1 Fig. Phylogenetic Trees of Four Bacterial Genera in condors.** Close-ups on reference tree of reads recruited for four representative samples, highlighting a region of the phylogenetic tree near (top-left) *Fusobacterium mortiferum*, (top-right) *Clostridium perfringens*, (bottom-left) *Lactobacillus johnsonii* and (bottom-right) *Propionibacterium avidum*. Selected reads from a phylogenetic marker gene (RNA Polymerase) were examined with Blastn or Blastp against the non-redundant databases at NCBI to identify the most closely related reference organism.
(PNG)

**S2 Fig. Results of 16S OTU analysis.** The fraction of the sample represented in the indicated OTU is shown as a heat map with a factor two ratio between adjacent colors, which run from dark blue to red. The rows are sorted phylogenetically, while the columns are in the same order as in Table 1 and Fig 1, but with the two samples for which the 16S samples failed left out of the heat map. The columns represent different samples in the same order as in Fig 2, and the rows represent the different nodes of the phylogeny. To the left of the heat map, we provide the most specific phylogenetic assignment QIIME made at the 80% confidence level, grouped phylogenetically in the same order as in Fig 2. These phylogenetic assignments correspond to the 16S sequences deposited in the Sequence Read Archive for this study. Representative 16S sequences are provided in S2 File, and are numbered according to the identifiers provided on the left of the heat map above. To the right of the heat map are the species names inferred from the 16S sequence and knowledge of what species were found with the RNA polymerase nucleotide reads identified by Sequedex from the shotgun sequencing data Similar to the process used to make Fig 2, representative sequences for each OTU were compared to the

Ribosomal DataBase project using BLASTN to identify the species provided to the right of the Figure.
(PNG)

**S3 Fig. The phylogeny of condors' microbiomes, comparing genes assembled from each sample to nearby genes from the Sequedex reference tree.** Phylogenetic tree of the RNA polymerase genes from reference genomes phylogenetically near to Delftia, and four assembled RNA polymerase genes from each of five mature cloacal samples. The sample numbers 'CCP.xx' refer to Table 1, while the '1' or '2' refer to forward and reverse reads, which were kept separate. Two equivalent assemblies were made to both the forward and reverse reads, and they are referred to as 'a' and 'b'. The alignment used to compute the tree is provided as a Supplementary text file. S3 Fig shows a phylogenetic tree resulting from such analysis.
(PNG)

**S4 Fig. Subsystems associated with stress response.** Selected subsystems distinctive of the stress response and efflux of toxic compounds are plotted, with the y-axis measuring the abundance of functionally identified reads associated with the particular subsystem, for each of the samples.
(TIF)

**S5 Fig. Subsystems associated with fecal metabolism.** Selected subsystems distinctive of the fecal microbiome are plotted, with the y-axis measuring the abundance of functionally identified reads associated with the particular subsystem, for each of the samples.
(TIF)

**S1 File. Spreadsheet with six tabs (a-f) of phylogenetic and functional classifications by various metrics. (a)** Phylogenetic rollup using Sequedex to classify shotgun metagenomic reads. **(b)** Normalized counts of reads assigned to each node of the Sequedex phylogenetic tree. **(c)** Operational taxonomic unit (OTU) counts for the 16S reads from each sample. **(d)** Functional rollup using Sequedes to classify shotgun metagenomic reads according to the high-level SEED classifications. **(e)** Normalized counts of shotgun metagenomic reads assigned to each SEED subsystem. **(f)** Normalized counts of shotgun metagenomic reads assigned to each Pfam family.
(XLSX)

**S2 File. Representative 16S sequences of OTUs, identified by the labels provided in Supplementary information S1c, provided in fasta format.**
(XLSX)

**S3 File. Representative genes assembled from open reading frames identified by Sequedex, in a spreadsheet format.**
(XLSX)

**S1 Text. Supplemental information text for additional descriptions for the figures and data files.**
(DOCX)

## Acknowledgments

We would like to thank T. Cade and R. Watson of the Peregrine Fund for their assistance in initiating this study and the Peregrine Fund for collection of fecal samples and swabs of the California condor.

## Author Contributions

**Conceptualization:** Benjamin H. McMahon, Jonathan Longmire, Nicolas W. Hengartner, Marti Jenkins, Jeanne M. Fair.

**Data curation:** Jonathan Longmire, Marti Jenkins, Jeanne M. Fair.

**Formal analysis:** Benjamin H. McMahon, Joel Berendzen, Nicolas W. Hengartner, Judith R. Cohn, Andrew W. Bartlow.

**Funding acquisition:** Benjamin H. McMahon, Jeanne M. Fair.

**Investigation:** Benjamin H. McMahon, Jonathan Longmire, Nicolas W. Hengartner, Jeanne M. Fair.

**Methodology:** Benjamin H. McMahon, Joel Berendzen, Cheryl Gleasner, Nicolas W. Hengartner, Momchilo Vuyisich, Judith R. Cohn.

**Project administration:** Benjamin H. McMahon, Jeanne M. Fair.

**Resources:** Benjamin H. McMahon, Jeanne M. Fair.

**Software:** Benjamin H. McMahon, Joel Berendzen, Nicolas W. Hengartner, Judith R. Cohn.

**Supervision:** Benjamin H. McMahon, Jeanne M. Fair.

**Visualization:** Benjamin H. McMahon, Nicolas W. Hengartner, Andrew W. Bartlow.

**Writing – original draft:** Lindsey Jacobs, Benjamin H. McMahon, Cheryl Gleasner, Andrew W. Bartlow, Jeanne M. Fair.

**Writing – review & editing:** Benjamin H. McMahon, Cheryl Gleasner, Andrew W. Bartlow, Jeanne M. Fair.

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
