## [Decision Letter · Decision Letter 0]

27 Jul 2019

PONE-D-19-18040

California Condor Microbiomes: Bacterial Variety and Functional Properties in Captive-Bred Individuals

PLOS ONE

Dear Dr. Fair,

Thank you for submitting your manuscript to PLOS ONE. After careful consideration, we feel that it has merit but does not fully meet PLOS ONE’s publication criteria as it currently stands. Therefore, we invite you to submit a revised version of the manuscript that addresses the points raised during the review process.

Both the reviewers and I think the manuscript has a lot of merit, although a number of changes were recommended.  In particular, there are some places in the manuscript that could be removed to shorten it, such as the description of sequencing in the introduction.  This may help to better focus the paper.  

We would appreciate receiving your revised manuscript by Sep 10 2019 11:59PM. To enhance the reproducibility of your results, we recommend that if applicable you deposit your laboratory protocols in protocols.io, where a protocol can be assigned its own identifier (DOI) such that it can be cited independently in the future. For instructions see: http://journals.plos.org/plosone/s/submission-guidelines#loc-laboratory-protocols

We look forward to receiving your revised manuscript.

Kind regards,

Suzanne L. Ishaq, PhD

Academic Editor

PLOS ONE

Journal Requirements:

1. Thank you for including your competing interests statement; "The authors have declared that no competing interests exist."

We note one ore more authors are affiliated to Viome, and GenerisBio.

Reviewers' comments:

Reviewer's Responses to Questions

**Comments to the Author**

1. Is the manuscript technically sound, and do the data support the conclusions?

Reviewer #1: Partly

Reviewer #2: Partly

2. Has the statistical analysis been performed appropriately and rigorously? 

Reviewer #1: Yes

Reviewer #2: Yes

3. Have the authors made all data underlying the findings in their manuscript fully available?

Reviewer #1: Yes

Reviewer #2: Yes

4. Is the manuscript presented in an intelligible fashion and written in standard English?

Reviewer #1: Yes

Reviewer #2: Yes

5. Review Comments to the Author

Reviewer #1: Dear authors,

I have had the pleasure of reviewing your paper for the second time, and want to thank you for taking my previous comments into account. The manuscript shows interesting and important results by comparing different sequencing methods in microbiomes of captive condors. I have a few overall comments, and below specific, line by line comments.

Overall, it is important to mention early in the manuscript that metagenomic sequencing represents potential functional profiles, and not functions actually occurring at the time of sampling. This is mentioned somewhere in the discussion, but should be moved up. Also, use of the word species for bacteria is complicated. If the authors used OTUs for clustering of 16S reads, they should be referred to as such.

I hope my comments help improve the manuscript.

Sincerely,

Kirsten Grond

kirsten.grond@uconn.edu

Introduction

The introduction is very thorough, which makes it long. If this is not a problem with the journal, leave it like this. If it is, you could possibly move part of the sequencing description in L114-126 to the methods section

Results:

L139. Add references or discuss this in the Discussion instead.

L168-170. Remove this sentence, as this information is in the figure description.

L172-174. Back this statement up with alpha diversity metrics & statistics.

L177-183. Can be moved to figure description.

L188-189. Remove this sentence

L198-202. Move to figure description

L204-206. Move to methods

L207. polymerase

L208-211. If matches are only to genus level, mention genus only and do not pick a well-known species instead (if I understood this right). Using a species that is potentially incorrect misrepresents results.

L212-216. The authors should not use rarefaction curves to say anything about diversity. Alpha diversity metrics with appropriate statistical tests should be used.

L217-220. Not necessary to describe what was done. Just report results.

L222. Test statistic for permanova is R2.

L222-224. Instead of using permanova for pairwise comparisons I would suggest using TukeyHSD. When eliminating a variable from permanovas/changing order of variables the results change.

L234-235. Seems more like a methodological discussion point than a result

L235-236. If the sequencing was deep enough (>100x coverage) and assembled, the authors can check the alignment rate of the raw reads back to the contigs or bins, and the alignment rate would tell if they are getting fine scaled phylogenies. For example, an alignment rate of >90% would be good and likely include most members of the community. A low alignment rate of <50% would indicate incomplete coverage and phylogenies will be missing a large portion of the communities.

L238-244. Move to methods.

L244-246. Move to figure caption or delete. Only report the results from the analyses here.

L249. Based on Delftia strain ID?

L251-254. I am unsure what the goal of this paragraph is. State the main findings of figure 4b instead?

L257-258. Remove sentence.

L262-263. Move this topic to discussion

L264-266. What do the authors mean by polarized extremes?

L271. Birds-eye! :D

L272-275. Move to figure description

L283-289. Seems more appropriate in methods. Also the first mention of the genes investigated, which should definitely be in the methods.

Discussion

The discussion is thorough, but very long. It contains several paragraphs that are part method or can be omitted. I indicated below which sections can potentially be removed.

L301. The authors should be consistent in terminology. Metagenomics is also used as a term for 16S in the manuscript, which could be confusing.

L302-304. Sentence needs clarification

L306-308. Do you know anything about Delftia genome size? If they have large genomes, that could also explain the difference in detection between the methods. Also, sequencing depth is much larger in the metagenomics.

L313-315. I don’t think this sentence is necessary.

L321-327. Good points!

L329-330. Some members of the classes are pathogenic, not the whole class.

L330-332. Merge these sentences to improve flow.

L333. Fusobacteria

L332-348. Super cool that physiology is so selective!

L364-365. Remove sentence

L371-373. This is a good thing to mention, but would fit better in a paragraph discussing the differences between fecal and cloacal samples.

L374-376. Would not report a math on 93% similarity. At most that is a genus level match.

L387-397. This describes methods and can be removed from the discussion

L398-403. Rewrite this paragraph with alpha diversity results.

L405-409. Can be removed

L414. Grained

L430-431. Remove sentence

L440-441. Remove sentence or explain the concepts.

L442-452. This would be useful to move to methods, where signatures are explained.

L455-463. This section should be in the Results as it states results but does not explain/discuss them.

L464-466. Methods

L467-478. This paragraph does not appear to add much to the discussion, and could be removed in the interest of shortening this section.

Conclusion

L480. Sentence reads funky.

L485-486. Use as first sentence of the paragraph and include main finding.

L500-502. Remove sentence. More samples are always needed, but the authors have a nice sample set here and the last sentence of the manuscript should not take away from that.

Methods

L542-543. Potential PCR bias in low DNA samples because of the different cycle numbers? Is cycle number a significant predictor for microbiome diversity/community composition?

L554-560. Add degree symbol for C

L569-570. Need a lot more information on analysis parameters here. Which programs were used for trimming, assembly, annotation, references etc. Did the authors align the raw reads to the TOL database and use those annotations for taxa? If so, there likely are a large number of false ID’s. To avoid this problem, the authors could assemble their reads, and align to that assembly. Any reads that don’t align to those contigs can be annotated for function separately, but should not be trusted for taxa info.

L600. Did the authors look if results changes using other distance matrices? Maybe add to supplement?

L606. Is Delftia a genus or species? Which genes did you look at?

L611. Expand the 16S analyses section. Which OTU-picking strategy and reference database did you use? What % was used for OTU cutoff?

Figures

Overall:

- Non-gradient colors would make the graph easier to read

Figure 1

- The labeling on the graph is confusing. There are places with two bacteria per color, and it is unclear whether this is a mistake or if the class is split between them. Would recommend to change the colors (or overlay with pattern) and add separate legend for the most abundant species.

- Add label to y-axis.

- In the caption it is mentioned that species were identified with among others16S data, which does not have the resolution for species identification.

Figure 2

- Why does not every row have something assigned to it? It is difficult to see which species corresponds with what data in the way this figure is presented. Also, expand the figure so full bacterial names are on one line and italicize them. I still don’t understand why the condor reads are in this figure. What does that add to the paper?

Figure 3

- Move to supplement. Not essential for main manuscript.

Figure 5A

- label y-axis

Figure 5C

- Move the legend to not overlap with the bars in the graph.

Figure 6

- increase symbol size and make text more readable.

Figure S3.

- What do the different colors mean?

Reviewer #2: To authors:

This paper utilizes two analyses methods to compare the microbiome of cloacal and fecal samples of captive California condors. The strengths of this text are 1. It provides baseline knowledge of how a scavenging bird’s microbiome looks like. 2. It demonstrates that the different analyses approaches can potentially produce dissimilar results. Simple and straightforward writing is not a strength of this text and authors should make an effort to improve the readability of the paper. I hope that this manuscript motivates further research into host-associated microbes in wild scavenger birds.

General comments:

1. My main and only big criticism is that you could have used additional variables to explain the observed patterns. It is well known that sex and breeding status con affect the microbiome. Is there a chance you could rerun some of the analyses to further explain your results including at least sex of the bird?

2. Your discussion would benefit if you provide further information on the bacterial transmission/colonization processes of these birds.

Specific comments. (L=line)

1. L43: Provide a broader conclusion

2. L49: Should be decade

3. Figure 1: Why are there random bacterial names on top of the bacterial groups? They make the figure harder to read.

4. L180-181: This belongs in the figure legend.

5. L199: Willing to make? I don’t understand.

6. Figure 3: Title is not necessary

7. Figure 3: Why are all the rarefaction curves cut at the same number of sampled sequences?

8. Figure 3: Could you provide the variance in the curves?

9. L258: delete the first are

10. L296-300: Can you provide hypotheses as to why immature condors have a more diverse microbiome?

11. L360-362: Could you provide a reference here?

6. PLOS authors have the option to publish the peer review history of their article (what does this mean?). If published, this will include your full peer review and any attached files.

Reviewer #1: Yes: Kirsten Grond

Reviewer #2: No

---

## [Author Response · Author response to Decision Letter 0]

22 Oct 2019

Responses to the specific and thoughtful review comments:

Introduction

The introduction is very thorough, which makes it long. If this is not a problem with the journal, leave it like this. If it is, you could possibly move part of the sequencing description in L114-126 to the methods section.

We worked to shorten the introduction to make it more concise, while trying to maintain the flow of the introduction for the work. 

Results:

L139. Add references or discuss this in the Discussion instead.

References have been added.

L168-170. Remove this sentence, as this information is in the figure description.

This sentence has been removed.

L172-174. Back this statement up with alpha diversity metrics & statistics. 

More statistics regarding species richness and true ‘alpha’ diversity have been added to the manuscript as suggested. For alpha diversity, we now refer to Simpson’s diversity and provide appropriate statistics.

L177-183. Can be moved to figure description.

These sentences have been moved to the figure description as suggested.

L188-189. Remove this sentence

This sentence has been removed.

L198-202. Move to figure description

These sentences have been moved to the figure description.

L204-206. Move to methods

This sentence is defining labels on figure S2, and has been moved to the caption for that figure.

L207. Polymerase

This misspelling has been corrected.

L208-211. If matches are only to genus level, mention genus only and do not pick a well-known species instead (if I understood this right). Using a species that is potentially incorrect misrepresents results. 

For many genera, the 16S sequences match to a small subset of the species in the genus. Rather than discard all of that information, we selected an interpretable representative of what matched, but warned the reader that the resolution was not always unique to a species, and provided the Fasta file keyed to the OTU representative sequence. It should also be noted that even when only one species from the database is matching, the sequence potentially matches other species, not in the database. So, we feel the added information from picking a representative species from among 2 or 3 (which was typical) offsets the information lost by simply providing the genus. Listing all matching species would be prohibitively cumbersome.

L212-216. The authors should not use rarefaction curves to say anything about diversity. Alpha diversity metrics with appropriate statistical tests should be used.

We reanalyzed the refraction curves and left Figure 3 in the main text as it seemed more useful with the additional confidence intervals around the curves. The rarefaction curves are computed by sampling the number of 16S sequences indicated on the x-axis and counting the number of distinct observed species, plotted on the y-axis. The shaded areas around the lines represent the 95% confidence intervals. None of the confidence intervals overlap, suggesting that each group significantly differs from the other two even when only 10,000 sequences are considered. In addition, it is more appropriate to state the rarefaction curves are for species richness and this was changed throughout the paper. 

L217-220. Not necessary to describe what was done. Just report results.

This section has been edited so that methods are not included.

L222. Test statistic for permanova is R2.

The test statistic has been corrected.

L222-224. Instead of using permanova for pairwise comparisons I would suggest using TukeyHSD. When eliminating a variable from permanovas/changing order of variables the results change. 

TukeyHSD will not work on distance matrices or on the output of NMDS analyses. The intent of these analyses is to survey microbial communities in different sample types to see if they differ in terms of their composition. Here, we follow the practice of microbial community ecology even though it may not be perfect. These pairwise PERMANOVAs are common in microbiome studies. We do not have to do pairwise comparisons since there is some resistance to such practices. If the reviewers would rather us not do pairwise comparisons, we would be happy to remove them. Doing the pairwise differences will tell us which groups are different from the others. Even though re-ordering the variables will change the specific results (compared to when all are included), we are still testing the hypothesis that each pair of groups is significantly different from one another. We compare our p-values to a corrected alpha level to avoid Type I errors.

L234-235. Seems more like a methodological discussion point than a result 

It could also fit in the methods, but where it is now it acts as a transition to a description of what our methods couldn’t do and what we did because of that lack.

L235-236. If the sequencing was deep enough (>100x coverage) and assembled, the authors can check the alignment rate of the raw reads back to the contigs or bins, and the alignment rate would tell if they are getting fine scaled phylogenies. For example, an alignment rate of >90% would be good and likely include most members of the community. A low alignment rate of <50% would indicate incomplete coverage and phylogenies will be missing a large portion of the communities. 

We agree this makes sense, but we did not want to get sidetracked assembling complex communities, only to see a slightly different qualitative heuristic. We did not think it made sense for us to assemble things, and we acknowledge it is qualitative. 

L238-244. Move to methods.

As suggested, these sentences have been moved to the methods.

L244-246. Move to figure caption or delete. Only report the results from the analyses here.

This has been moved to the figure caption.

L249. Based on Delftia strain ID?

Based on where microbiome reads for each bird fall within the phylogenetic tree – some explanation has also been added to the sentence.

L251-254. I am unsure what the goal of this paragraph is. State the main findings of figure 4b instead?

This paragraph serves as a transition from the phylogeny-based analysis to a function-based analysis. Figure 4B is mentioned because the two data types are distinct in that figure.

L257-258. Remove sentence.

This sentence has been removed.

L262-263. Move this topic to discussion

As suggested, this topic has been moved to the discussion.

L264-266. What do the authors mean by polarized extremes?

The polarized extremes portion of the sentence has been removed.

L271. Birds-eye! :D

L272-275. Move to figure description

This information has been moved to the figure description.

L283-289. Seems more appropriate in methods. Also, the first mention of the genes investigated, which should definitely be in the methods.

As suggested, this has been moved to the methods.

Discussion

The discussion is thorough, but very long. It contains several paragraphs that are part method or can be omitted. I indicated below which sections can potentially be removed.

L301. The authors should be consistent in terminology. Metagenomics is also used as a term for 16S in the manuscript, which could be confusing.

The manuscript has been read through and edits made to keep metagenomics and 16S as separate, consistent terms.

L302-304. Sentence needs clarification

This sentence has been removed.

L306-308. Do you know anything about Delftia genome size? If they have large genomes, that could also explain the difference in detection between the methods. Also, sequencing depth is much larger in the metagenomics. 

Delftia acidovorans does have a large genome (6.9 mega-bases) but it also has five copies of the ribosome, which is above average for bacteria. Both 16S and metagenomic analysis are by-read fractions, so the ratio of Delftia to others will be independent of sequencing depth. Also, Delftia is <3% of the sample based on the OUT analysis (supplementary table 1) but is 80% of the sample by metagenomics (Figure 1). This ratio is much larger than is explainable by genome size or ribosomal copy number effects.

L313-315. I don’t think this sentence is necessary.

This sentence has been removed. 

L321-327. Good points!

L329-330. Some members of the classes are pathogenic, not the whole class.

Good point. The wording has been edited.

L330-332. Merge these sentences to improve flow.

As suggested, the sentences have been merged. 

L333. Fusobacteria

This word has been corrected.

L332-348. Super cool that physiology is so selective!

L364-365. Remove sentence

This sentence has been removed.

L371-373. This is a good thing to mention, but would fit better in a paragraph discussing the differences between fecal and cloacal samples.

A paragraph mentioning differences between fecal and cloacal samples has been added.

L374-376. Would not report a math on 93% similarity. At most that is a genus level match. 

Here we are trying to show that some of our unknown organisms are close to reference genomes. However, you are correct that it is misleading in how it was written. It has been edited for clarity.

L387-397. This describes methods and can be removed from the discussion

This paragraph was removed from the paper.

L398-403. Rewrite this paragraph with alpha diversity results. 

We rewrote this paragraph to address the new results we did for species richness. 

L405-409. Can be removed

This section has been removed.

L414. Grained

This word has been corrected.

L430-431. Remove sentence

This sentence has been removed.

L440-441. Remove sentence or explain the concepts. 

Added Ben’s description and a reference to Berendzen et al. 2012

L442-452. This would be useful to move to methods, where signatures are explained.

This has been moved to methods, as suggested.

L455-463. This section should be in the Results as it states results but does not explain/discuss them.

Good point. This section has been moved to Results.

L464-466. Methods

This sentence has been moved to methods.

L467-478. This paragraph does not appear to add much to the discussion, and could be removed in the interest of shortening this section.

As suggested, this paragraph has been removed.

Conclusion

L480. Sentence reads funky.

The sentence has been edited to read better.

L485-486. Use as first sentence of the paragraph and include main finding.

This sentence has been used as the first sentence of the paragraph and has been modified to include our main findings.

L500-502. Remove sentence. More samples are always needed, but the authors have a nice sample set here and the last sentence of the manuscript should not take away from that.

This sentence has been removed.

Methods

L542-543. Potential PCR bias in low DNA samples because of the different cycle numbers? Is cycle number a significant predictor for microbiome diversity/community composition? 

We looked at if cycle number was related in any way to the results for the microbiome communities. All of the samples were run together. However, with a few of the samples, there was less DNA extracted than the majority of the samples, which had ample DNA. 

L554-560. Add degree symbol for C

A degree symbol has been added.

L569-570. Need a lot more information on analysis parameters here. Which programs were used for trimming, assembly, annotation, references etc. Did the authors align the raw reads to the TOL database and use those annotations for taxa? If so, there likely are a large number of false ID’s. To avoid this problem, the authors could assemble their reads, and align to that assembly. Any reads that don’t align to those contigs can be annotated for function separately, but should not be trusted for taxa info. 

We used Sequedex extensively. Sequedex avoids all of the complexities and sources of error mentioned by the reviewer. We will add another sentence to the paragraph mentioned to clear up confusion by briefly describing Sequedex and pointing the reader to an extensive description of it (Berendzen et al. 2012).

L600. Did the authors look if results changes using other distance matrices? Maybe add to supplement?

Bray-Curtis distances are the most robust for community data and are commonly used for microbiome studies. We ran the NMDS and PERMANOVAs again with other types of distance matrices and they all showed the same results. We do not see a need to add additional figures to supplemental data.

L606. Is Delftia a genus or species? Which genes did you look at? 

Delftia is a genus. We looked at the genes in two Delftia reference genomes, one for Delftia acidovorans, and the other for Delftia spp., where the reference was not assigned a species name.

L611. Expand the 16S analyses section. Which OTU-picking strategy and reference database did you use? What % was used for OTU cutoff? 

We addressed this concern in the text of the paper. 

The 16S data analysis was performed with QIIME version 1.8 [42] with representative 16S sequences are provided in Supplementary Information SI2, and were selected using the QIIME utility, single_rarefaction.py, and a value for a sequence similarity parameter of 0.94. Annotations of 16S sequences were made by comparing to the 97_otus.fasta database.

Figures

Overall:

- Non-gradient colors would make the graph easier to read

For each of the figure comments below, we work to improve the figure quality per the recommendations. 

Figure 1

- The labeling on the graph is confusing. There are places with two bacteria per color, and it is unclear whether this is a mistake or if the class is split between them. Would recommend to change the colors (or overlay with pattern) and add separate legend for the most abundant species.

- Add label t o y-axis.

- In the caption it is mentioned that species were identified with among others16S data, which does not have the resolution for species identification.

Figure 2

- Why does not every row have something assigned to it? It is difficult to see which species corresponds with what data in the way this figure is presented. Also, expand the figure so full bacterial names are on one line and italicize them. I still don’t understand why the condor reads are in this figure. What does that add to the paper?

Figure 3

- Move to supplement. Not essential for main manuscript.

We actually redid this rarefaction analysis and curve and added confidence intervals. We left his figure in the main section as we felt it added more value now with the new analysis and confidence intervals. 

Figure 5A

- label y-axis

Figure 5C

- Move the legend to not overlap with the bars in the graph.

Figure 6

- increase symbol size and make text more readable.

We increased the font size and readability of this figure. We increase the symbol size but then they obstructed the view of the other symbols and labels. Instead, we focused on making the overall figure better in the overall presentation. 

Figure S3.

- What do the different colors mean? 

Reviewer #2: To authors:

This paper utilizes two analyses methods to compare the microbiome of cloacal and fecal samples of captive California condors. The strengths of this text are 1. It provides baseline knowledge of how a scavenging bird’s microbiome looks like. 2. It demonstrates that the different analyses approaches can potentially produce dissimilar results. Simple and straightforward writing is not a strength of this text and authors should make an effort to improve the readability of the paper. I hope that this manuscript motivates further research into host-associated microbes in wild scavenger birds.

General comments:

1. My main and only big criticism is that you could have used additional variables to explain the observed patterns. It is well known that sex and breeding status con affect the microbiome. Is there a chance you could rerun some of the analyses to further explain your results including at least sex of the bird? 

Yes, we agree that there are other variables that may explain some observed patterns. However, regarding sex, the number of males and females are very unbalanced. Because there are 3 types (groups) of samples, we would have to do a test of sex 3 times. The sample sizes for females are way too low compared to males for each group. For example, there are 9 male fecal samples compared to only 1 female. For cloacal immature samples, 12 are males and only 3 are females. Because of this unbalanced design, we chose not to run statistics to avoid mistaken inferences.

2. Your discussion would benefit if you provide further information on the bacterial transmission/colonization processes of these birds. 

We added text to describe the living condition and co-habitation of the birds. Due to the movement of the birds and openness of the facility, this opportunity is as close as you can get to a controlled experiment for raising birds together with the same exact diet in a laboratory. Due to the openness of the facility, no birds were considered isolated and therefore, have different exposure to microbes than other birds. 

Specific comments. (L=line)

1. L43: Provide a broader conclusion 

This was also mentioned by the other reviewer. We now state our two conclusions in the paper, which are 1) We characterized gut microbiota from healthy captive California condors to facilitate the design of future studies 2) demonstrate novel methods that can be applied to other systems

2. L49: Should be decade

This word has been corrected.

3. Figure 1: Why are there random bacterial names on top of the bacterial groups? They make the figure harder to read. 

The bacterial names are explained in the figure caption: most of the phyla are dominated by a couple species, and these species were listed to aid in interpretation.

4. L180-181: This belongs in the figure legend.

This description has been moved to the figure legend.

5. L199: Willing to make? I don’t understand.

The wording of this sentence has been edited to remove the personification.

6. Figure 3: Title is not necessary 

We deleted this figure as suggested. 

7. Figure 3: Why are all the rarefaction curves cut at the same number of sampled sequences? This was because this was all done to roughly the same depth of sequencing, and statistical significance drops at some point. 

8. Figure 3: Could you provide the variance in the curves? 

Yes, we added confidence intervals to this figure and they added some additional support in the figure to show how the three groups did indeed differ. Adding confidence intervals was one of the reasons we left the figure in the primary section. 

9. L258: delete the first are

The entire sentence has been removed.

10. L296-300: Can you provide hypotheses as to why immature condors have a more diverse microbiome? 

This is definitely one of the most interesting results of the study since the few previous studies of the development of the microbiome in birds have reported the opposite. Without additional studies, our thinking and hypothesis is that it takes some time for biological communities to compete with each other and settle down in this species. This may be due to the more acidic environment of the condors compared to other species. We added some text in the discussion as it is an important point to bring up for the scientific community to potentially investigate in the future. 

11. L360-362: Could you provide a reference here?

As suggested, references have been provided.

---

## [Decision Letter · Decision Letter 1]

14 Nov 2019

California Condor Microbiomes: Bacterial Variety and Functional Properties in Captive-Bred Individuals

PONE-D-19-18040R1

Dear Dr. Fair,

We are pleased to inform you that your manuscript has been judged scientifically suitable for publication and will be formally accepted for publication once it complies with all outstanding technical requirements.

With kind regards,

Suzanne L. Ishaq, PhD

Academic Editor

PLOS ONE

Additional Editor Comments (optional):

Reviewers' comments:

Reviewer's Responses to Questions

**Comments to the Author**

1. If the authors have adequately addressed your comments raised in a previous round of review and you feel that this manuscript is now acceptable for publication, you may indicate that here to bypass the “Comments to the Author” section, enter your conflict of interest statement in the “Confidential to Editor” section, and submit your "Accept" recommendation.

Reviewer #1: All comments have been addressed

Reviewer #2: All comments have been addressed

2. Is the manuscript technically sound, and do the data support the conclusions?

Reviewer #1: Yes

Reviewer #2: Yes

3. Has the statistical analysis been performed appropriately and rigorously? 

Reviewer #1: Yes

Reviewer #2: Yes

4. Have the authors made all data underlying the findings in their manuscript fully available?

Reviewer #1: Yes

Reviewer #2: Yes

5. Is the manuscript presented in an intelligible fashion and written in standard English?

Reviewer #1: Yes

Reviewer #2: Yes

6. Review Comments to the Author

Reviewer #1: Overall, I am happy with how the authors have addressed my comments. I have reviewed this manuscript for a different journal before reviewing it for PLoS, and the authors have done a great job at addressing all comments. Please congratulate them from me on their effort and a great manuscript.

Reviewer #2: (No Response)

7. PLOS authors have the option to publish the peer review history of their article (what does this mean?). If published, this will include your full peer review and any attached files.

Reviewer #1: Yes: Kirsten Grond

Reviewer #2: No

---

## [Editor Report · Acceptance letter]

19 Nov 2019

PONE-D-19-18040R1 

California Condor Microbiomes: Bacterial Variety and Functional Properties in Captive-Bred Individuals 

Dear Dr. Fair:

I am pleased to inform you that your manuscript has been deemed suitable for publication in PLOS ONE. Congratulations! Your manuscript is now with our production department. 

With kind regards,

on behalf of

Dr. Suzanne L. Ishaq 

Academic Editor

PLOS ONE